# Towards Real-time Video Compressive Sensing on Mobile Devices

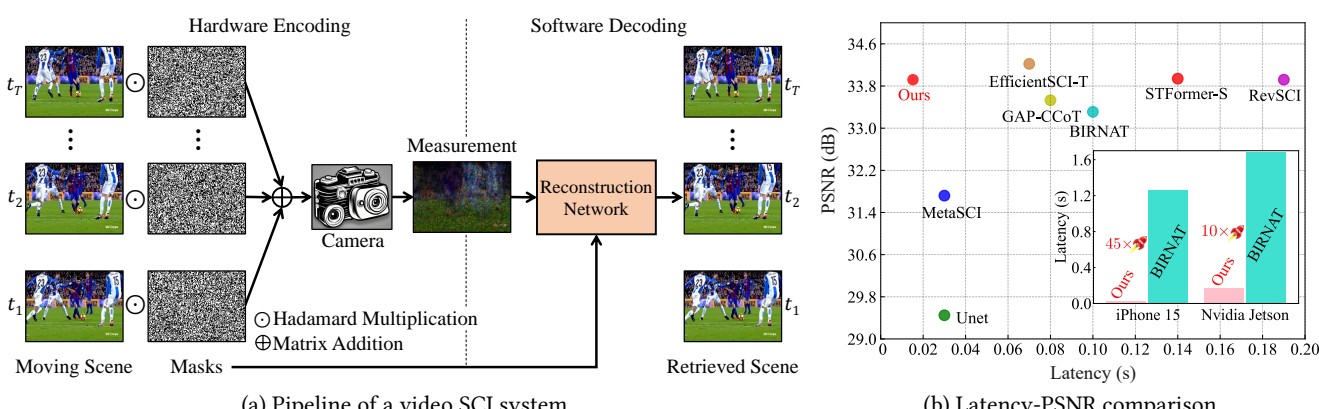

(a) Pipeline of a video SCI system  (b) Latency-PSNR comparison

Figure 1: (a) Illustration of the video Snapshot Compressive Imaging (SCI) system: In the hardware encoding process, a high-speed scene is modulated by different masks and then the modulated scene is captured by a low-speed 2D camera as snapshot measurements. In the software decoding process, the captured measurements and the corresponding masks are fed into a reconstruction algorithm to retrieve the desired high-speed video frames. (b) Comparison of the reconstruction quality and running speed of different video SCI reconstruction methods on the mobile devices and the NVIDIA GPU platform. Our proposed MobileSCI network can achieve state-of-the-art reconstruction quality with real-time performance.

## ABSTRACT

Video Snapshot Compressive Imaging (SCI) uses a low-speed 2D camera to capture high-speed scenes as snapshot compressed measurements, followed by a reconstruction algorithm to retrieve the high-speed video frames. The fast evolving mobile devices and existing high-performance video SCI reconstruction algorithms motivate us to develop mobile reconstruction methods for real-world applications. Yet, it is still challenging to deploy previous reconstruction algorithms on mobile devices due to the complex inference process, let alone real-time mobile reconstruction. To the best of our knowledge, there is no video SCI reconstruction model designed to run on the mobile devices. Towards this end, in this paper, we present an effective approach for video SCI reconstruction, *dubbed MobileSCI*, which can run at *real-time* speed on mobile devices for the first time. Specifically, we first build a U-shaped 2D convolution-based architecture, which is much more efficient and mobile-friendly than previous state-of-the-art reconstruction methods. Besides, an efficient feature mixing block, based on the channel splitting and shuffling mechanisms, is introduced as a novel bottleneck block of our proposed MobileSCI to alleviate the computational burden. Finally, a customized knowledge distillation strategy is utilized to further improve the reconstruction quality. Extensive results on both simulated and real data show that our proposed MobileSCI can achieve superior reconstruction quality with high efficiency on the mobile devices. Particularly, we can reconstruct a $256 \times 256 \times 8$ snapshot compressed measurement with real-time performance (**about 35 FPS**) on an iPhone 15. Code of this paper will be released.

## CCS CONCEPTS

• **Computing methodologies → Reconstruction**.

## KEYWORDS

Computational imaging, Snapshot compressive imaging, Mobile system, Real-time reconstruction, Mobile network

## 1 INTRODUCTION

To record high-speed scenes, researchers usually rely on high-speed cameras which suffer from high hardware cost and require large transmission bandwidth. Inspired by compressed sensing [7], video Snapshot Compressive Imaging (SCI) provides a promising solution as it can capture high-speed scenes using a low-speed 2D camera with low bandwidth. As shown in Fig. 1(a), there are two main stages in a video SCI system: hardware encoding and software decoding [35]. In the hardware encoding process, we first modulate the high-speed scene with different random binary masks, and then the modulated scene is compressed into a series of snapshot measurements which are finally captured by a low-cost and low-speed 2D camera. So far, many successful video SCI hardware encoders [6, 9, 21, 39] have been built. In the software decoding stage, the captured snapshot measurements and the modulation

masks are fed into a reconstruction algorithm to retrieve the desired high-speed video frames. In this aspect, numerous video SCI reconstruction algorithms [4, 5, 28, 30–32], mostly based on the deep neural networks, have been proposed with superior reconstruction quality. Therefore, it is time to consider the real-world applications of the video SCI system.

Nowadays, mobile devices such as smartphones or embedded devices are fast evolving, which inspires us to seek the possibility of implementing a mobile video SCI system. When considering the mobile deployment, the biggest challenge is the limited computational resources in the mobile platforms. However, the inference of previous video SCI reconstruction algorithms such as BIRNAT [5], RevSCI [4], STFormer [27], and EfficientSCI [26] is usually complex. In other words, it is difficult to deploy previous video SCI reconstruction algorithms on the low-end mobile devices. Therefore, existing video SCI reconstruction methods run on server-side, resulting in dependency on Internet connection, speed, and server usage costs for the potential users of the video SCI system.

To the best of our knowledge, mobile video SCI reconstruction has never been explored within the research community. In this paper, we propose the first video SCI reconstruction network to be deployed on the mobile devices, *dubbed MobileSCI* which is computationally more efficient than previous state-of-the-art (SOTA) deep learning-based reconstruction methods. Specifically, we first revisit the network design of previous video SCI reconstruction methods to identify the unfriendly operations when deploying on the mobile devices. Particularly, we find that complex operations such as deep unfolding inference and 3D convolutional layers are the primary computational bottlenecks hindering the mobile deployment of previous video SCI reconstruction methods. Bearing these in mind, we build a U-shaped 2D convolution-based architecture which is much more efficient and mobile-friendly than previous SOTA reconstruction methods. Besides, to reduce model size and computational complexity and improve network capability of our proposed MobileSCI model, we introduce the channel splitting and shuffling mechanisms to build an efficient feature mixing block as a novel bottleneck block. Finally, knowledge distillation by a stronger teacher network also improves the reconstruction quality of the proposed MobileSCI. As shown in Fig. 1(b), our MobileSCI-KD network can achieve comparable reconstruction quality with much faster inference speed than previous SOTA reconstruction methods on the mobile devices as well as the NVIDIA GPU platform. In particular, our MobileSCI-KD runs 45× and 10× faster than previous SOTA BIRNAT on an iPhone 15 and a NVIDIA Jetson Orin Nano platform, respectively.

Our contributions can be summarized as follows:

- We propose *MobileSCI*, to the best of our knowledge, *the first mobile video SCI reconstruction network*.
- A U-shaped architecture is built with computational efficient and mobile-friendly 2D convolutional layers. Following this, an efficient feature mixing block, based on the channel splitting and shuffling mechanisms, is introduced as a novel bottleneck block of our proposed MobileSCI to further alleviate the computational burden.
- We implement a customized knowledge distillation strategy which further improves the reconstruction quality.

- Comprehensive experimental results show that our MobileSCI network can achieve superior performance with much better real-time performance than previous SOTA methods, especially on the mobile devices.

## 2 RELATED WORK

### 2.1 Video SCI Reconstruction Methods

Current video SCI reconstruction methods can be divided into *traditional iteration-based methods* and *deep learning-based ones*. The traditional iteration-based methods formulate the video SCI reconstruction process as an optimization problem with regularization terms such as total variation [34] and Gaussian mixture model [33], and solve it via iterative algorithms. The major drawback of these iteration-based methods is their time-consuming iterative optimization process. To improve the running speed, Yuan *et al.* propose to plug a pre-trained denoising model into each iteration of the optimization process [36, 37]. However, it still take a long time to reconstruct large-scale scenes.

Recently, researchers begin to utilize deep neural networks in the video SCI reconstruction task. For example, BIRNAT [5] uses a bidirectional recurrent neural network to exploit the temporal correlation. MetaSCI [30] is designed to solve the fast adaptation problem of video SCI by introducing a meta modulated convolutional network. RevSCI [4] builds the first end-to-end 3D convolutional neural network, and adopt the reversible structure to save model training memory. Wang *et al.* build the first Transformer-based video SCI reconstruction method [27] with space-time factorization and local self-attention mechanism. After that, Wang *et al.* develop an efficient reconstruction network [26] based on dense connections and space-time factorization. Combining the idea of iteration-based methods and deep learning-based ones, several deep unfolding networks [31, 32, 40] are proposed.

Due to their superior performance, we favor the deep learning-based reconstruction methods. However, it is challenging to deploy them on the mobile devices due to the existence of the complex operations. Therefore, in this paper, we mainly focus on developing *mobile-friendly video SCI reconstruction algorithm* with careful network design.

### 2.2 Mobile Networks

So far, a variety of computational efficient architectures have been proposed which are more suitable for mobile deployment. The first well-known architectures are the MobileNet family. Among them, MobileNet V1 [11] first introduces the depth-wise separable convolutional layers as a basic unit to build lightweight deep neural networks. Following this, MobileNet V2 [22] proposes the inverted residual structure which is more memory efficient. Finally, MobileNetV3 [10] is tuned to mobile devices through a combination of hardware-aware network architecture search complemented by the NetAdapt algorithm. After that, numerous MobileNet Variants [2, 8, 24, 25] have been proposed with superior performance. Another emerging branch is the mobile Transformer design. For example, EdgeViTs [18] introduces a highly cost-effective local-global-local information exchange bottleneck, for the first time, enable attention based vision models to compete with the SOTA light-weight CNNs. Chen *et al.* leverage the advantages of both

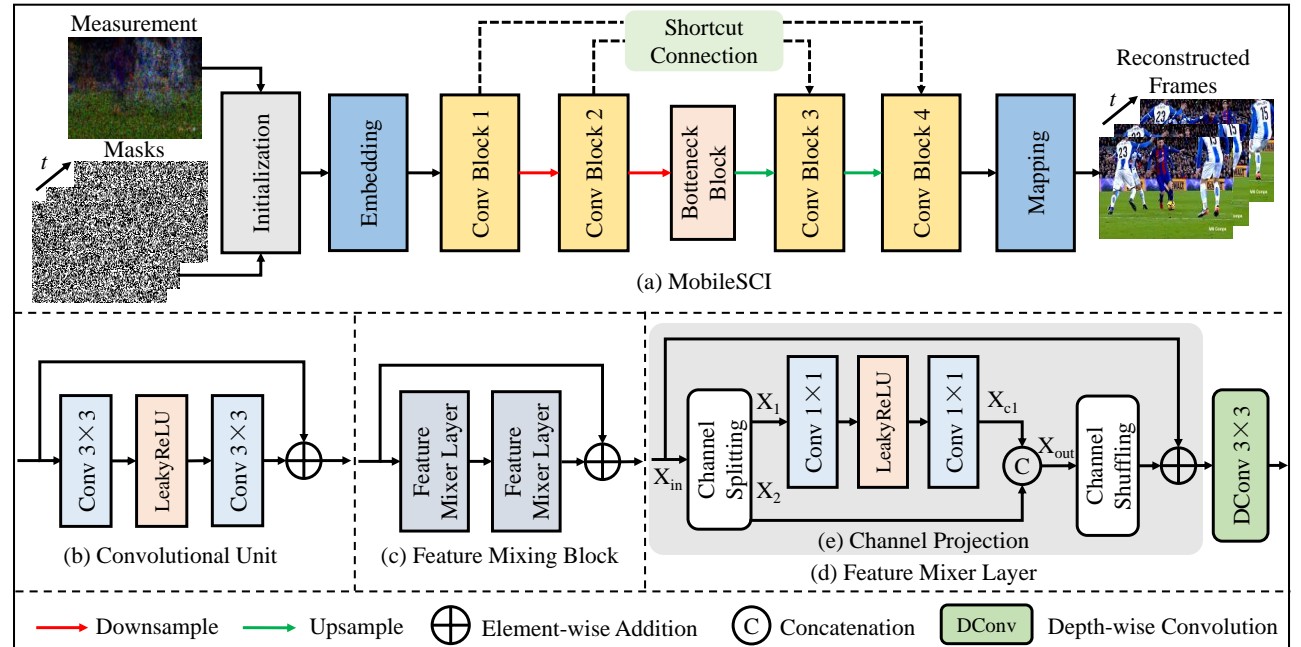

**Figure 2: (a) Overall pipeline of the proposed MobileSCI network. It is a 2D convolution-based encoder-decoder architecture. The encoder consists of two convolutional blocks and two downsample modules. The bottleneck is a convolutional block or an efficient feature mixing block. The decoder contains two convolutional blocks and two upsample modules. Each convolutional block contains several convolutional units. (b) The convolutional unit contains two $3 \times 3$ convolutional layers followed by a LeakyReLU function. (c) The feature mixing block is composed of two feature mixer layers. (d) The feature mixer layer composes of a channel projection layer and a $3 \times 3$ depth-wise convolutional layer. (e) In the channel projection layer, we first split the input feature $\mathrm{X}_{in}$ along the channel dimension as $\mathrm{X}_1$ and $\mathrm{X}_2$. Then, $\mathrm{X}_1$ undergoes two $1 \times 1$ convolutional layer followed by a LeakyReLU function to get the output feature $\mathrm{X}_{c1}$. Next, we concatenate $\mathrm{X}_{c1}$ and $\mathrm{X}_2$ to obtain the output feature. Finally, a channel shuffling operation is implemented. Note that, to better propagate the extracted features, we add some shortcut connections in the proposed MobileSCI network.**

CNN and Transformer with a parallel design of MobileNet and Transformer with a two-way bridge in between [3]. By rethinking the mobile block design for efficient attention-based networks, [14] and [38] present two novel lightweight vision Transformer architectures: EfficientFormer and EMO.

The success of the above-mentioned mobile architectures and the fast evolving mobile devices motivate us to deploy the video SCI reconstruction algorithms on the mobile devices. Therefore, in this paper, *we are the first* to explore the mobile-friendly network design for video SCI reconstruction.

## 3 PRELIMINARY: VIDEO SCI SYSTEM

Here, we describe the forward model of the video SCI system. Let $\{\mathbf{X}_t\}_{t=1}^{T} \in \mathbb{R}^{n_x \times n_y}$ denotes a $T$-frame high-speed scene to be captured in a single exposure time, where $n_x, n_y$ represent the spatial resolution of each frame and $T$ is the compression ratio (Cr) of the video SCI system. Following this, the modulation process can be modeled as multiplying $\{\mathbf{X}_t\}_{t=1}^{T}$ by pre-defined masks $\{\mathbf{M}_t\}_{t=1}^{T} \in \mathbb{R}^{n_x \times n_y}$, that is $\mathbf{Y}_t = \mathbf{X}_t \odot \mathbf{M}_t$, where $\{\mathbf{Y}_t\}_{t=1}^{T} \in \mathbb{R}^{n_x \times n_y}$ and $\odot$ denote the modulated frames and Hadamard (element-wise) multiplication, respectively. Finally, a low-speed 2D camera is used to capture the modulated high-speed scene as a snapshot compressed measurement $\mathbf{Y} \in \mathbb{R}^{n_x \times n_y}$. Thus, the forward process of

the video SCI system can be fomulated as,

$$\mathbf{Y} = \sum_{t=1}^{T} \mathbf{X}_t \odot \mathbf{M}_t + \mathbf{N}, \qquad (1)$$

where $\mathbf{N} \in \mathbb{R}^{n_x \times n_y}$ denotes the noise.

In the decoding stage of the video SCI system, we can get the desired video frames $\{\hat{\mathbf{X}}_t\}_{t=1}^{T}$ by feeding the compressed measurement $\mathbf{Y}$ and the modulation masks $\{\mathbf{M}_t\}_{t=1}^{T}$ into a video SCI reconstruction network.

## 4 OUR PROPOSED METHODS

### 4.1 Motivation

In this section, we rethink previous video SCI reconstruction algorithm designs from a mobile efficiency perspective. First of all, iteration-based reconstruction methods are not applicable for mobile deployment due to their time-consuming property. Secondly, the deep unfolding framework is not feasible for the mobile devices due to the complex deep unfolding inference procedure. Finally, we turn our attention to the end-to-end (E2E) deep neural network-based reconstruction methods. On the one hand, several 2D convolution-based methods including MetaSCI [30] and U-net [20] enjoy high efficiency. However, the reconstruction quality is unsatisfactory (lower than $32dB$). On the other hand, the

E2E video SCI reconstruction methods such as RevSCI [4] and EfficientSCI [26], can achieve superior reconstruction quality (higher than $33dB$). However, they are build upon 3D convolutional layers or the non-local self-attention modules which are computational heavy and thus not friendly for the mobile devices. Now, we take EfficientSCI [26] as an example. It can present the best efficiency performance along with high reconstruction quality when testing on the NVIDIA GPU platform. However, due to the memory constrain, we cannot deploy EfficientSCI on the mobile devices. Base on the above analysis, we believe that an E2E deep neural network with 2D convolutional layers shall be a mobile-friendly and efficient video SCI reconstruction algorithm design.

## 4.2 Overall MobileSCI Architecture

The overall architecture of our MobileSCI is shown in Fig. 2(a). In the initialization stage, inspired by [5, 27], we use the estimation module to pre-process measurement ($\mathbf{Y}$) and masks ($\mathbf{M}$) as follows,

$$\overline{\mathbf{Y}} = \mathbf{Y} \oslash \sum_{t=1}^{T} \mathbf{M}_t, \; \mathbf{X}_e = \overline{\mathbf{Y}} \odot \mathbf{M} + \overline{\mathbf{Y}}, \qquad (2)$$

where $\oslash$ represents Hadamard (element-wise) division, $\overline{\mathbf{Y}} \in \mathbb{R}^{n_x \times n_y}$ is the normalized measurement, which preserves a certain degree of the background and motion trajectory information, and $\mathbf{X}_e \in \mathbb{R}^{n_x \times n_y \times T}$ represents the coarse estimate of the desired video. We then take $\mathbf{X}_e$ as the input of the proposed network to get the final reconstruction result.

Our proposed MobileSCI network is mainly composed of three parts: $i$) feature extraction module, $ii$) feature enhancement module, and $iii$) video reconstruction module. Firstly, the feature extraction module is composed of a 2D convolutional layers with a kernel size of $3 \times 3$, followed by a LeakyReLU activation function [17]. With the proposed feature extraction module, we can effectively project the input video frames into the high-dimensional feature space and produce the shallow features. After that, inspired by CST [1] which has been successfully applied on the hyperspectral SCI reconstruction task [13], we adopt a U-shaped structure in the feature enhancement module of the MobileSCI model to generate the deep features. Finally, the video reconstruction module, composed of a 2D convolutional layer with a kernel size of $3 \times 3$, is able to conduct video reconstruction on the deep features output by the feature enhancement module.

## 4.3 U-shaped Feature Enhancement Module

The proposed U-shaped feature enhancement module consists of an encoder $\mathcal{E}$, a bottleneck $\mathcal{B}$ and a decoder $\mathcal{D}$. Among them, $\mathcal{E}$ consists of two convolutional blocks and two downsample modules. The downsample module is a strided $3 \times 3$ convolutional layer which can downscale the feature maps and double the channel dimension. $\mathcal{B}$ is a convolutional block or an efficient feature mixing block. $\mathcal{D}$ contains two convolutional blocks and two upsample modules. The upsample module is a $1 \times 1$ convolutional layer followed by a PixelShuffle operation [23]. In each convolutional block, we stack several convolutional units. As shown in Fig. 2(b), the convolutional unit is a residual style module which is composed of two $3 \times 3$ convolutional layers followed by a LeakyReLU activation function. Note that, the channel number of the features keeps consistent in

the convolutional unit. Finally, to alleviate the information loss during rescaling, we add some shortcut connections between the encoder $\mathcal{E}$ and the decoder $\mathcal{D}$.

## 4.4 Efficient Feature Mixing Block

To further enhance our MobileSCI, we aim to reduce its model size and alleviate its computational burden. The key challenges are two-folds: $i$) Mobile attention modules [14, 38] have achieved superior performance on multiple vision tasks. However, when applying attention modules on the extracted features with large spatial size, the real-time performance will be largely worsen. $ii$) The inverted residual structure [22] is not feasible for the U-shaped architecture because channel number of the bottleneck layers is not the constrain for improving the reconstruction quality. Bearing the above in mind, we propose an efficient feature mixing block, based on the channel splitting and shuffling mechanisms [16], as a novel bottleneck structure of the proposed MobileSCI. As shown in Fig. 2(c), our proposed feature mixing block is composed of two feature mixer layers with a shortcut connection.

**Feature Mixer Layer:** As shown Fig. 2(d),The feature mixer layer contains a channel projection layers and a $3 \times 3$ depth-wise convolutional layer. To mix features at the channel locations, we adopt point-wise MLPs to perform channel projection. Besides, we introduce the channel splitting and shuffling mechanisms to reduce computational cost in the channel projection layer. Specifically, as shown Fig. 2(e), in each channel projection layer, we first split the input feature $\mathbf{X}_{in}$ along the channel dimension as $\mathbf{X}_1$ and $\mathbf{X}_2$. Then, $\mathbf{X}_1$ undergoes two point-wise MLPs followed by a LeakyReLU function to generate the output feature $\mathbf{X}_{c1}$. Among them, the first MLP expands the channel number twice, and the second MLP reduces the channel number by half. Next, we concatenate $\mathbf{X}_{c1}$ and $\mathbf{X}_2$ to obtain the output feature. Finally, a channel shuffling operation is employed to enable the information exchange on the concatenated feature. The above procedure of the channel projection layer can be expressed as follows,

$$\begin{aligned} [\mathbf{X}_1, \mathbf{X}_2] &= \text{Split}(\mathbf{X}_{in}), \\ \mathbf{X}_{c1} &= \mathbf{W}_2(\sigma(\mathbf{W}_1(\mathbf{X}_1))), \qquad (3) \\ \mathbf{X}_{out} &= \text{Shuffle}(\text{Concat}(\mathbf{X}_{c1}, \mathbf{X}_2)) + \mathbf{X}_{in}, \end{aligned}$$

where $\text{Split}(\cdot)$ and $\text{Shuffle}(\cdot)$ denote the splitting and shuffling of features in the channel dimension. $\mathbf{W}_1$ and $\mathbf{W}_2$ are the point-wise convolutional layers. $\sigma$ represents the LeakyReLU function. $\text{Concat}(\cdot)$ is the concatenation operation.

## 4.5 Loss Function

As established in Sec. 3, our proposed method takes the measurement ($\mathbf{Y}$) and the corresponding masks ($\{\mathbf{M}_t\}_{t=1}^{T}$) as inputs, and then generates the dynamic video frames ($\{\hat{\mathbf{X}}_t\}_{t=1}^{T} \in \mathbb{R}^{n_x \times n_y}$). Given the ground truth ($\{\mathbf{X}_t\}_{t=1}^{T} \in \mathbb{R}^{n_x \times n_y}$), we choose the mean square error (MSE) as our loss function, which can be expressed as,

$$\mathcal{L}_{MSE} = \frac{1}{T n_x n_y} \sum_{t=1}^{T} \|\hat{\mathbf{X}}_t - \mathbf{X}_t\|_2^2. \qquad (4)$$

For the knowledge distillation process, we choose a deeper MobileSCI model as the teacher network due to its strong representation ability. Then we put a knowledge-distillation loss between the

teacher output $\{\hat{X}_k\}_{k=1}^T \in \mathbb{R}^{n_x \times n_y}$ and student output $\hat{X}_t$ as,

$$\mathcal{L}_{KD} = \frac{1}{T n_x n_y} \sum_{t=1}^{T} \|\hat{X}_k - \hat{X}_t\|_2^2. \tag{5}$$

Therefore, the final loss function for our MobileSCI network will be a weighted combination:

$$\mathcal{L} = \lambda_1 \mathcal{L}_{MSE} + \lambda_2 \mathcal{L}_{KD}, \tag{6}$$

where $\lambda_1 = 1, \lambda_2 = 1$ in our experimental settings.

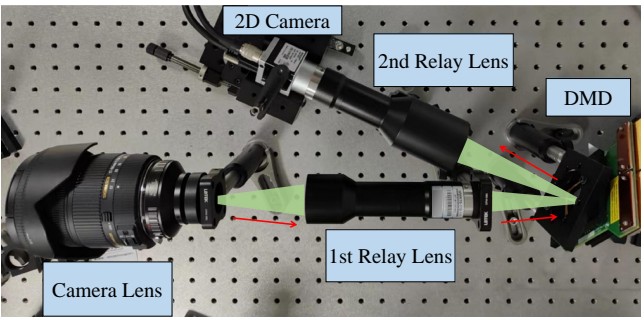

**Figure 3: Illustration of the real built video SCI system.**

## 5 EXPERIMENTS

**Hardware Implementation:** The optical setup of the real video SCI system is shown in Fig. 3. The encoding process can be summarized as follows: First, the reflected light from the target scene is imaged onto the surface of the digital micromirror device (DMD) (TI, $2560 \times 1600$ pixels, $7.6 \mu m$ pixel pitch) via a camera lens (Sigma, 17-50/2.8, EX DC OS HSM) and the first relay lens (Coolens, WWK10-110-111). Then, the projected dynamic scene is modulated by the random binary masks loaded in the DMD. Finally, the encoded scene is projected onto a low-speed 2D camera (Basler acA1920, $1920 \times 1200$ pixels, $4.8 \mu m$ pixel pitch) with the second relay lens (Coolens, WWK066-110-110), which is captured in a snapshot manner. In our experiments, the camera works at 50 FPS, thus the equivalent sampling rate of the real built video SCI system is $50 \times$ Cr FPS.

**Real Data Acquisition Procedure:** The first essential step of the real data acquisition process is to record the modulated masks in a right way. Specifically, we first place a Lambertian white board at the objective plane of the camera lens. Then, each mask pattern ($M^0$) which is projected on the surface of DMD, will be recorded sequentially. Moreover, to eliminate the influence of background noise and nonuniform light distribution, we need to capture two extra images corresponding to a pure white ($O$ with all '1's) and black mask ($Z$ with all '0's), respectively. Finally, we can get the actual mask ($M$) with (7).

$$M = (M^0 - Z) \oslash (O - Z), \tag{7}$$

where $\oslash$ stands for Hadamard (element-wise) division.

**Training and Testing Datasets:** Following BIRNAT [5], we choose DAVIS2017 [19] with resolution $480 \times 894$ as the training dataset. For the testing dataset, we first test our MobileSCI network on six simulated testing data (including Kobe, Traffic, Runner, Drop, Crash and Aerial with a size of $256 \times 256 \times 8$) to verify the model

performance. After that, we test our MobileSCI model on the real testing data (including Water Balloon and Domino with a size of $512 \times 512 \times 10$) captured by the real video SCI system described in the "Hardware Implementation" section.

**Training Details:** Following DeSCI [15], eight sequential frames are modulated by a set of eight random binary masks and then summed up to get a series of snapshot compressed measurements as described in Sec. 3. We randomly crop patch cubes ($256 \times 256 \times 8$) from original scenes in DAVIS2017 and obtain 26,000 training data pairs with data augmentation operations including random scaling and random flipping. Following this, we adopt Adam [12] to optimize the model with an initial learning rate of 0.0001. After iterating for 100 epochs on the training data with a $128 \times 128$ resolution and 20 epochs on the training data with a $256 \times 256$ resolution, we adjust the learning rate to 0.00001 and continue to iterate for 20 epochs on the training data with a $256 \times 256$ resolution to obtain the final model parameters. The whole training process is conducted on 4 NVIDIA RTX 3090 GPUs based on PyTorch 1.13.1.

### 5.1 Results on Simulated Data

We compare our proposed MobileSCI network with the traditional iteration-based methods (GAP-TV [34], PnP-FFDNet [36], PnP-FastDVDnet [37], and DeSCI [15]) and the deep learning-based ones (U-net [20], MetaSCI [30], BIRNAT [5], RevSCI [4], GAP-CCoT [28], STFormer-S [27], and EfficientSCI-T [26]) on the simulated testing datasets. In this section, the peak-signal-to-noise-ratio (PSNR) and structured similarity index metrics (SSIM) [29] are used as the evaluation metrics of the reconstruction quality. Running time on the GPU platform is adopted as the evaluation metrics of the algorithm efficiency. Moreover, for a fair comparison, we test all the deep learning-based reconstruction methods on the same NVIDIA RTX 3090 GPU. We customize the MobileSCI model by setting the channel number of the embedding output feature, the number of the convolutional unit in each convolutional block and the number of the feature mixing block in the bottleneck block as 64, 6, and 1, respectively. Additionally, the final MobileSCI-KD model is obtained by distilling from a stronger network.

We can observe from Tab. 1 that our proposed MobileSCI-KD can achieve much better real-time performance with comparable reconstruction quality than previous SOTA methods. In particular, i) The PSNR value of our MobileSCI-KD surpasses U-net and MetaSCI by a large margin on average ($4.47dB$ and $2.20dB$, respectively), while reducing the testing time by more than $2.0\times$. ii) The PSNR value of our MobileSCI-KD model surpasses BIRNAT and GAP-CCoT by $0.61dB$ and $0.39dB$ on average, while reducing the testing time by about $6.7\times$ and $5.3\times$, respectively. iii) Our MobileSCI-KD can achieve comparable reconstruction quality with RevSCI and STFormer-S, while the testing time is reduced by more than $12.7\times$ and $9.3\times$, respectively. iv) Although EfficientSCI-T outperforms our MobileSCI-KD by about $0.3dB$, it required more than $4.7\times$ longer inference time on the NVIDIA RTX 3090 GPU. Besides, EfficientSCI-T cannot be deployed on the mobile devices.

For visualization purposes, we plot several reconstructed video frames in Fig. 4, where we can see from the zooming areas in each selected video frame that our MobileSCI-KD can provide comparable high-quality reconstructed images with previous SOTA STFormer-S, and EfficientSCI-T. Especially for the complex Traffic, Crash

**Table 1: The average PSNR in dB (left entry), SSIM (right entry) and running time per measurement of different video SCI reconstruction algorithms on 6 simulated testing datasets. Note that, for a fair comparison, all the experiments are conducted on the same NVIDIA GPU.**

| Method | Kobe | Traffic | Runner | Drop | Crash | Aerial | Average | Latency (s) |
|---|---|---|---|---|---|---|---|---|
| GAP-TV [34] | 26.46, 0.885 | 20.89, 0.715 | 28.52, 0.909 | 34.63, 0.970 | 24.82, 0.838 | 25.05, 0.828 | 26.73, 0.858 | 4.20 (CPU) |
| PnP-FFDNet [36] | 30.50, 0.926 | 24.18, 0.828 | 32.15, 0.933 | 40.70, 0.989 | 25.42, 0.849 | 25.27, 0.829 | 29.70, 0.892 | 3.00 (GPU) |
| PnP-FastDVDnet [37] | 32.73, 0.947 | 27.95, 0.932 | 36.29, 0.962 | 41.82, 0.989 | 27.32, 0.925 | 27.98, 0.897 | 32.35, 0.942 | 6.00 (GPU) |
| DeSCI [15] | 33.25, 0.952 | 28.71, 0.925 | 38.48, 0.969 | 43.10, 0.993 | 27.04, 0.909 | 25.33, 0.860 | 32.65, 0.935 | 6180.00 (CPU) |
| U-net [20] | 27.79, 0.807 | 24.62, 0.840 | 34.12, 0.947 | 36.56, 0.949 | 26.43, 0.882 | 27.18, 0.869 | 29.45, 0.882 | 0.03 (GPU) |
| MetaSCI [30] | 30.12, 0.907 | 26.95, 0.888 | 37.02, 0.967 | 40.61, 0.985 | 27.33, 0.906 | 28.31, 0.904 | 31.72, 0.926 | 0.03 (GPU) |
| BIRNAT [5] | 32.71, 0.950 | 29.33, 0.942 | 38.70, 0.976 | 42.28, 0.992 | 27.84, 0.927 | 28.99, 0.917 | 33.31, 0.951 | 0.10 (GPU) |
| RevSCI [4] | 33.72, 0.957 | 30.02, 0.949 | 39.40, 0.977 | 42.93, 0.992 | 28.12, 0.937 | 29.35, 0.924 | 33.92, 0.956 | 0.19 (GPU) |
| GAP-CCoT [28] | 32.58, 0.949 | 29.03, 0.938 | 39.12, 0.980 | 42.54, 0.992 | 28.52, 0.941 | 29.40, 0.923 | 33.53, 0.958 | 0.08 (GPU) |
| STFormer-S [27] | 33.19, 0.955 | 29.19, 0.941 | 39.00, 0.979 | 42.84, 0.992 | 29.26, 0.950 | 30.13, 0.934 | 33.94, 0.958 | 0.14 (GPU) |
| EfficientSCI-T [26] | 33.45, 0.960 | 29.20, 0.942 | 39.51, 0.981 | 43.56, 0.993 | 29.27, 0.954 | 30.32, 0.937 | 34.22, 0.961 | 0.07 (GPU) |
| MobileSCI-KD (Ours) | 32.28, 0.946 | 28.91, 0.936 | 39.51, 0.981 | 43.13, 0.992 | 29.39, 0.953 | 30.32, 0.936 | 33.92, 0.957 | 0.015 (GPU) |

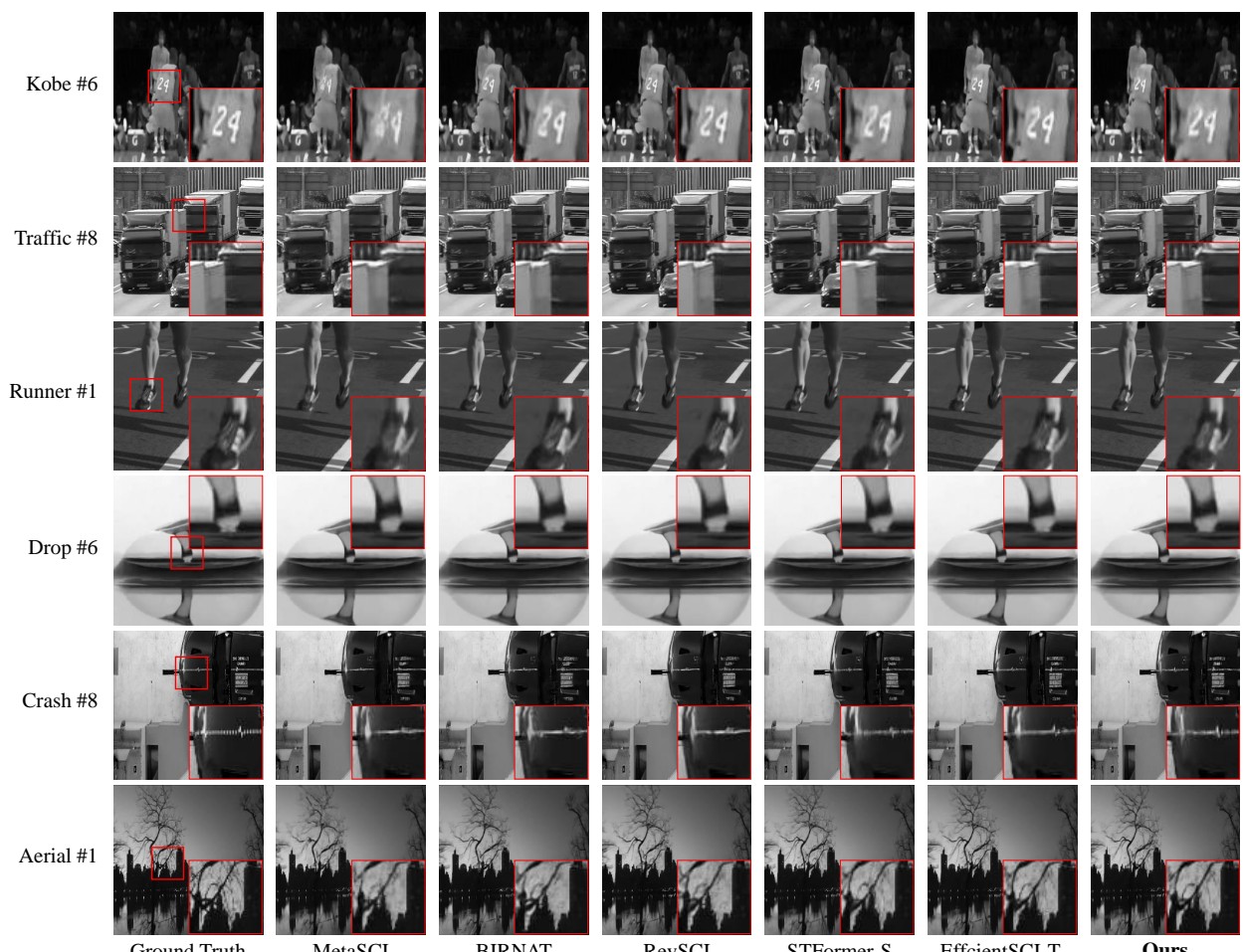

**Figure 4: Selected reconstruction video frames of the simulated testing datasets. For a better view, we zoom in on a local area as shown in the small red boxes of each ground truth image, and do not show the small red boxes again for simplicity.**

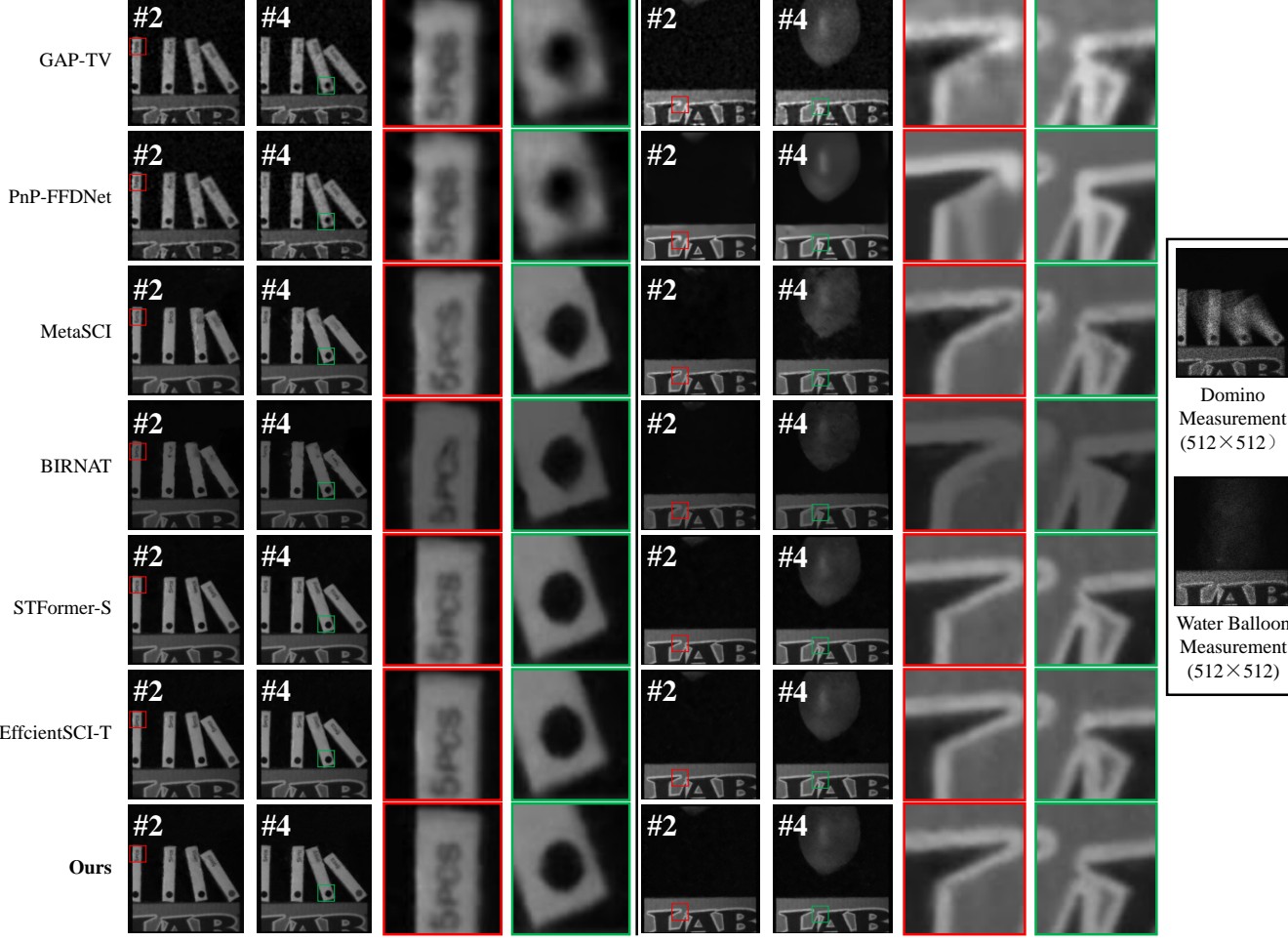

**Figure 5: Selected reconstruction video frames of the real data. For a better view, we zoom in on two local areas as shown in the small red and green boxes of the first and second image.**

and Aerial scenes, we can observe sharp edges and more details in the reconstructed frames of our proposed MobileSCI-KD.

**Table 2: The actual speed of our MobileSCI-KD model and BIRNAT deployed on various mobile devices. The actual speed is measured in seconds. Some numbers are missing, since BIRNAT was providing an out of memory error.**

| Mobile Device | MobileSCI-KD Latency (s) | BIRNAT Latency (s) |
|---|---|---|
| iPhone 15 | 0.028 | 1.26 |
| iPad Pro (2nd gen) | 0.080 | 3.49 |
| OnePlus 11 | 1.31 | - |
| Xiaomi 14Ultra | 1.19 | - |
| NVIDIA Jetson Orin Nano | 0.17 | 1.68 |

We also measure the actual inference speed of our MobileSCI model on real mobile devices with input resolution as $256 \times 256 \times 8$. Due to the extensive use of 3D convolutional layers or Transformer modules, which consume a large amount of memory, some of the previous SOTA methods including RevSCI, GAP-CCoT, STFormer-S, and EfficientSCI-T cannot be deployed to resource-limited mobile

devices such as mobile phones and embedded devices. Thus, we compare our MobileSCI model with previous SOTA BIRNAT in this section. As shown in Tab. 2, our MobileSCI is more than 40× and 10× faster than BIRNAT on the mobile phones and NVIDIA embedded device, respectively. Particularly, our MobileSCI can achieve real-time (35 FPS) reconstruction on an iPhone 15.

## 5.2 Results on Real Data

We further test our proposed MobileSCI on the real data captured by the video SCI system. Here, it is worth mentioning that due to the mismatch between the camera and DMD, the real captured masks cannot meet the binary distribution as that of the simulated random binary masks. Thus, we need to finetune the model which is pretrained on the simulated training data. The training data pairs for finetuning are generated with the original scenes in DAVIS2017 and the real captured masks. In the finetuning process, we first set the learning rate as 0.0001. Then we can finetune for 20 epochs on the training data with a $128 \times 128$ resolution, followed by 10 epochs on the training data with a $256 \times 256$ resolution.

We can see from Fig. 5 that our proposed MobileSCI-KD can achieve comparable reconstruction quality with previous SOTA STFormer-S, and EfficientSCI-T. Specifically, our proposed MobileSCI-KD model can provide clearly reconstructed letters on the `Domino` data and sharp edges on the `Water Balloon` data.

## 5.3 Ablation Study

**Effect of the feature mixing block:** To evaluate the effectiveness of various efficient modules in our MobileSCI model, we conduct experiments on the efficient modules including the mobile attention module proposed in EfficientFormer V2 [14], the inverted residual structure shown in MobileNet V2 [22] and our proposed feature mixing block. In the baseline model, we adopt the convolutional unit (shown in Fig. 2(b)) in the whole U-shaped architecture. Then, we replace the convolutional unit in the bottleneck block with the proposed feature mixing block to get our MobileSCI-Base model.

We can get the following observations from Tab. 3 that: i) EfficientFormer V2 outperforms the baseline model with slightly $0.04dB$, while the inference time increases by about 2×. ii) MobileNet V2 can help reduce the model size and slightly alleviate the computational burden. However, the reconstruction quality drops by more than $0.17dB$ with no improvement on the real-time performance. iii) Our proposed feature mixing block is extremely lightweight and more efficient. Specifically, compared with the baseline model, the model size and computational complexity of our MobileSCI-Base model reduce by 2.14×, 18%, along with a 7.4% improvement on the inference speed. Moreover, the reconstruction quality is guaranteed. Thus, our proposed feature mixing block can achieve the best trade-off between accuracy and efficiency.

**Table 3: Ablation study on the feature mixing block. The latency on an iPhone 15 is reported.**

| Method | PSNR | SSIM | Params (M) | FLOPs (G) | Latency (ms) |
|---|---|---|---|---|---|
| Baseline | 33.78 | 0.956 | 12.052 | 149.05 | 30.25 |
| EfficientFormer V2 [14] | 33.82 | 0.956 | 12.207 | 142.38 | 60.15 |
| MobileNet V2 [22] | 33.61 | 0.955 | 8.186 | 133.17 | 30.50 |
| MobileSCI-Base (Ours) | 33.77 | 0.956 | 5.632 | 122.76 | 28.23 |

Then, we replace the convolutional units at different parts of the U-shaped architecture with the proposed feature mixing block. Keeping the same testing time on the same NVIDIA RTX 3090 GPU, we sequentially replace the convolutional units at the bottleneck block ("Case 1"), Conv Block 2&3 ("Case 2") and Conv Block 1&4 ("Case 3") of our MobileSCI model. As shown in Tab. 4, "Case 1" outperforms "Case 2" and "Case 3" by about $0.23dB$ and $1.04dB$, respectively. Therefore, replacing the convolutional units only at the bottleneck block can better ensure the reconstruction quality.

**Effect of knowledge distillation:** In this section, we study different knowledge distillation strategies. The teacher MobileSCI network is obtained by setting the channel number of the embedding output feature, the number of the convolutional unit in each convolutional block and the number of the feature mixing block in the bottleneck block as 64, 24 and 4, respectively. We first implement knowledge distillation on the randomly initialized student model ("Strategy 1"). However, we can see from Fig. 6 that this brings no performance improvement on the proposed MobileSCI-Base.

**Table 4: Ablation study on replacing the convolutional unit at different parts of our mobileSCI network. We sequentially replace the convolutional units at the Bottleneck Block (BB), Conv Block 2&3 (CB23) and Conv Block 1&4 (CB14) of our MobileSCI model.**

| Replacing Blocks | PSNR | SSIM | Params (M) | FLOPs (G) |
|---|---|---|---|---|
| BB | 33.77 | 0.956 | 5.632 | 122.76 |
| BB + CB23 | 33.54 | 0.954 | 3.758 | 92.00 |
| BB + CB23 + CB14 | 32.73 | 0.944 | 4.200 | 56.06 |

Therefore, we customize a novel knowledge distillation strategy ("Strategy 2"). Specifically, we select one layer every four layers from the teacher model to initialize the student model. Following this, we conduct knowledge distillation on the initialized student model. As shown in Fig. 6, "Strategy 2" brings $0.15dB$ PSNR improvement of the reconstruction quality with much faster convergence speed than "Strategy 1".

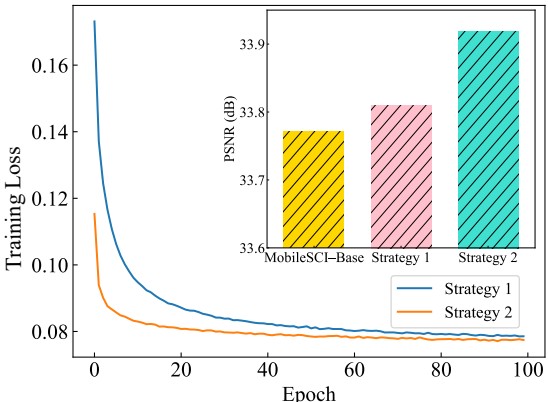

**Figure 6: Loss curve and PSNR value of different knowledge distillation strategies. In "Strategy 1", we implement knowledge distillation on the randomly initialized student model. In "Strategy 2", we first initialize the student model with selective layers from the teacher model. Then, we conduct knowledge distillation on the initialized student model.**

## 6 CONCLUSION

In this paper, we propose the first real-time mobile video SCI reconstruction method, dubbed *MobileSCI*, with effective network design. Specifically, a U-shaped architecture with 2D convolutional layers is built as the baseline. Following this, an efficient feature mixing block based on the channel splitting and shuffling mechanisms, is proposed to reduce computational cost and improve network capability of our proposed MobileSCI. Finally, a customized knowledge distillation strategy is introduced to further improve the reconstruction quality. Extensive experiments on both simulated and real testing data show that our proposed MobileSCI can achieve a good trade-off between accuracy and efficiency. More importantly, combining the proposed optical setup with our MobileSCI network, we contribute a promising way to build a whole mobile video SCI system with real-time performance.

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
