# OpenReview forum: "Towards Real-time Video Compressive Sensing on Mobile Devices"
_acmmm.org/ACMMM/2024/Conference — MM2024 Poster_

### Official Review · Reviewer_EByW · 2024-05-16

**Rating:** 4
**Confidence:** 3

**Summary:**

In this work, a light-weight video Snapshot Compressive Imaging (SCI) method is proposed to achieve real time SCI reconstruction on mobile devices. A U-shaped 2D convolution-based architecture with 2D convolutional layers is adopted as the baseline. A feature-mixture model with channel splitting and shuffling mechanisms is proposed to reduce computational cost and improve network capacity. Knowledge distillation strategy is introduced to further improve the performance.

**Strengths:**

* The performance is good, the proposed method achieves comparable reconstruction quality with lower time consumption.
* The acquisition of the real SCI data is well designed and demonstrated. The authors could consider opening the collected real test data.
* The running speed comparison on mobile device is innovative, it could be instructive if the more test details could be opened.

**Limitations:**

* Novelty is somehow limited. The architecture of the network is common for low complexity applications and no specific network design/discussion neither insights oriented for SCI task are discussed in the manuscript. The superiority of the method compared to other low-complexity SCI methods, e.g., MetaSCI and UNet is not explained, neither. And the knowledge distillation is also a common strategy in deep learning methods.
* In table 3, the number of parameters of the bottleneck layer in the baseline method seems to occupy an excessive proportion. This is counterintuitive and makes the effectiveness of the proposed feature mixing block on complexity reduction less convincing.
* The writing of the manuscript is somehow slapdash and the review suggests using more polishing.

**Suitability:**

3

---

### Official Review · Reviewer_qWWa · 2024-05-24

**Rating:** 6
**Confidence:** 4

**Summary:**

The paper primarily introduces MobileSCI, a real-time mobile video compression-aware sensing and reconstruction network based on the Spatial Compression Imaging (SCI) technique. This is the first application of video SCI technology on mobile devices. MobileSCI achieves real-time high frame rate video reconstruction on mobile devices while maintaining high reconstruction quality by optimizing the network structure, utilizing 2D convolutional layers, and employing an efficient feature fusion block based on channel splitting and shuffling mechanisms. The reconstruction performance is further improved through knowledge distillation strategy. MobileSCI achieves real-time performance, approximately 35 FPS, on mobile devices such as iPhone 15, surpassing existing methods.

**Strengths:**

1. The accomplishment of real-time generation on mobile devices, as demonstrated in this article, is truly remarkable. Implementing inference on mobile devices poses significant challenges that require careful optimization and consideration of resource constraints.
2. The experimental section showcases the performance of MobileSCI on both simulated and real-world data. Compared to existing methods, MobileSCI achieves significant improvements in real-time performance while maintaining similar reconstruction quality. Particularly on mobile devices, MobileSCI outperforms BIRNAT by a wide margin, demonstrating its capability for real-time video compression-aware sensing and reconstruction on resource-constrained devices.

**Limitations:**

1. In this article, when I saw the U-Shape network, the first thing that came to my mind was the structure of U-Net. But there is no discussion between them. The difference between the U-Shape 2D convolutional network to U-Net in this article needs to be emphasized more effectively.
2. Writing needs to be improve, but it's ok to read for now.

**Suitability:**

3

---

### Official Review · Reviewer_NeMP · 2024-05-26

**Rating:** 2
**Confidence:** 3

**Summary:**

In this paper, the authors propose the first mobile-friendly method for Video Compressive Sensing. The authors propose to use a UNet architecture with a shuffleNet block in the bottleneck layer for efficient video compressive sensing. The proposed method is 45x faster than previous state-of-the-art EfficientSCI-T while achieving comparable results. Moreover, the authors have shown that the proposed method can run on an iPhone 15 at 30 FPS.

**Strengths:**

1) This paper proposes a mobile-friendly approach for video compressive sensing that is 45x faster than the earlier state-of-the-art method (EfficientSCI-T)
2) The authors have implemented the proposed method on several mobile devices and have benchmarked the execution time. The proposed method can run at 30 FPS on iPhone 15
3) The authors have also experimented with other efficient architectures such as MobileNetV2 and shown that using ShuffleNet-based architecture yields the best results

**Limitations:**

1) The authors use a standard UNet architecture with a feature mixing block bottleneck layer. However, the proposed feature mixing block is a standard ShuffleNet unit. The use of knowledge distillation for efficient execution is also a known technique. Hence, the novelty of this paper is weak.

2) In Fig.5 for the water balloon scenario, the authors have shown slight improvement in a static region. However, the main purpose of compressive sensing is to recover regions with fast motion (i.e. the balloon region). Hence the authors should ideally show examples where improvements can be observed in regions with fast motion

3) L573: "EfficientSCI-T cannot be deployed on mobile devices" - Can you elaborate more on this?

4) Can you show a peak memory analysis on a mobile device against RevSCI, which is a memory-efficient method? For efficient on-device execution, both execution time and memory need to be analyzed.

**Suitability:**

2

---

### Official Review · Reviewer_unqb · 2024-05-29

**Rating:** 2
**Confidence:** 3

**Summary:**

The paper aims to deploy the above pipeline in mobile phones (here iPhone 15). However, the main problem is related to the reconstruction network. Some currently available reconstruction networks cannot be deployed on mobile phones, and if they do, there are issues related to reconstruction quality, latency, or memory.

The paper proposes a MobileSCI reconstruction network which is much more efficient. The innovation is done in three steps:

(i) A U-shaped architecture is built with 2D convolutional layers (**mobile friendly).
(ii) A feature mixing block is introduced into the architecture to reduce computational burden and model size.
(iii) The knowledge distillation strategy is utilized to improve the reconstruction quality further.

**Strengths:**

**The results are shown for iPhone 15, which I understand is not a low-end mobile device. The mobile phone has sufficient resources.

**Bottleneck: The term is introduced by the ResNet paper. It refers to a procedure where we reduce the number of channels, apply the convolution, and restore the channels again.

**The paper claims it is the first to explore the mobile-friendly network design for video SCI reconstruction.

The network has three components:

(i) Feature extraction module
(ii) Feature enhancement module
(iii) Video reconstruction module

Paper’s main component: Feature enhancement module!!

U-shaped Feature Enhancement Module

Components: Encoder, Bottleneck, and Decoder

Encoder: Two convolutional blocks and two downsampling modules (reduce dimension and double the channel dimensions)

Decoder: Two convolutional blocks and two upsampling modules

**Limitations:**

**From the results on the synthetic dataset, it is clear that in terms of latency, MobileSCI-KD outperforms every current SCI model on the NVIDIA GPU 3090.
Only the BIRNAT SCI model is compared on iPhone 15 with the MobileSCI-KD since we cannot deploy other SCI models on mobile phones (large memory need).

**No quantitative analysis is shown for testing on real data. Only qualitative analysis is shown. Qualitatively it is showing comparable performance with other SCI models.


**After analyzing the RevSCI and EfficientSCI, I found the architecture of MobileSCI is slightly different due to the:

(i) 2D convolution blocks are there while the earlier papers have 3D convolutional blocks. The feature mixing block is itself present in the above EfficientSCI paper, but the only difference is that MobileSCI uses 2D convolutional units.

**The paper should address why the earlier methods were using the 3D convolutional blocks how they found that 2D convolutional blocks are more efficient and what will be the side effects of ignoring the 3D convolutional blocks.

(ii) The knowledge distillation strategy is famous. Here, it is used to improve the reconstruction quality. But no novelty is found.

**Further, the entire analysis provided by the paper has already been done. The paper is incomplete in terms of giving details about working reasons and mathematical equations for the efficiency of the proposed feature mixing block.
The paper should have to include a great analysis of the only thing it is proposing.

**The supplementary material has no analysis. I think they have submitted it just for the formality.

**Suitability:**

2

---

### Meta-Review · Area_Chair_udoT · 2024-07-08

**Recommendation:** Accept (Poster)
**Confidence:** 3

**Metareview:**

The paper received mixed reviews, but the rebuttal helped reviewers better understand the paper and its contributions. Based on the reviews, the rebuttal, and updates, it is leaning more towards acceptance than rejection. The paper focuses on "efficient on-device execution" using state-of-the-art techniques and optimizations. While it doesn't propose a new technique, it optimizes existing ones, and it is believed that this work deserves space at ACM MM and would lead to valuable discussions.